# NT-proBNP Concentrations in the Umbilical Cord and Serum of Term Neonates: A Systematic Review and Meta-Analysis

**DOI:** 10.3390/diagnostics12061416

**Published:** 2022-06-08

**Authors:** Evangelos Christou, Zoe Iliodromiti, Abraham Pouliakis, Rozeta Sokou, Matina Zantiotou, Chrisa Petropoulou, Theodora Boutsikou, Nicoletta Iacovidou

**Affiliations:** 1Neonatal Department, Aretaieio Hospital, National and Kapodistrian University of Athens, 115 28 Athens, Greece; ziliodromiti@yahoo.gr (Z.I.); sokourozeta@yahoo.gr (R.S.); chrisapetra@gmail.com (C.P.); theobtsk@gmail.com (T.B.); niciac58@gmail.com (N.I.); 2PICU, Panagiotis and Aglaia Kyriakou Children’s Hospital, 115 27 Athens, Greece; 32nd Department of Pathology, “ATTIKON” University Hospital, National and Kapodistrian University of Athens, 124 61 Athens, Greece; apou1967@gmail.com; 4Children’s Hospital Agia Sofia, 115 27 Athens, Greece; matina.zantiotou@gmail.com

**Keywords:** biomarker, neonate, Nt-proBnp, serum, umbilical cord

## Abstract

The detection of NT-proBNP levels both in umbilical cord blood (UCB) samples and in serum samples collected from healthy term neonates during the neonatal period. A systematic review of relevant literature in accordance with PRISMA guidelines was conducted. For quality appraisal, the potential risk of bias was assessed using the BIOCROSS evaluation tool. The random-effects and fixed-effects models were used to calculate weighted mean differences with a corresponding 95% confidence interval. A total of forty (40) studies met the inclusion criteria for the systematic review. After further examination, eighteen (18) studies (1738 participants) from the UCB sample group and fourteen (14) studies (393 participants) from the serum sample group were selected to perform a meta-analysis. Using the fixed-effects model, the mean intervals of NT-proBNP in UCB and serum samples were 492 pg/mL (95% CI: 480–503 pg/mL) and 1341 pg/mL (95% CI: 1286–1397 pg/mL), respectively. A higher concentration of ΝΤ-proBNP was observed in the serum sample group compared to the UCB samples (*p* < 0.001). We present the intervals of NT-proBNP in UCB and in the serum of healthy term neonates. The determination of the potential effect of perinatal factors on the biomarker’s reference range was also aimed.

## 1. Introduction

Ever since Sudoh [1] discovered a peptide homologous to the ANP peptide isolated from porcine brain tissue in 1988, which subsequently was detected to be produced in the cardiac ventricles, BNP has been extensively studied.

The human BNP gene is located on chromosome 1 and encodes the 108-amino-acid precursor, proBNP. The prohormone is proteolytically processed by the enzyme furin and is cleaved into a biologically active 32-amino-acid peptide (BNP), after being separated from its biologically inert N-terminal fragment (NT-proBNP).

Cardiac myocytes constitute the major source of the neurohormone’s production, whereas cardiac fibroblasts, whose contribution is not yet accurately determined, are also considered responsible for its production. Although the atrium is the main cardiac production site under normal conditions, in the case of chronic myocardial stress, activation of ventricular hormone synthesis occurs. Furthermore, in contrast to ANP, atrial cells provide less storage for BNP, which demonstrates a lower basal production rate. BNP is characterized by an acute response to stress stimuli, in terms of a faster production onset and significantly higher plasma concentrations in patients with heart failure.

Natriuretic peptides are activated and exhibit their full biological activity upon binding to their respective biologically active receptors, namely NPR-A and NPR-B (also known as biologically active guanylyl cyclase receptors) and NPR-C, whose primary function is the clearance of the natriuretic peptides. Since natriuretic peptide receptors are located on most target organs involved in the cardiovascular homeostasis, the physiological effects of BNP are manifold. In the kidney, BNP increases the glomerular filtration rate and inhibits the tubular fractional reabsorption of sodium and water. In the cardiovascular system, it primarily demonstrates an autocrine function by releasing NO, resulting in the decrease in peripheral vascular resistance by relaxing the vascular smooth muscle cells. Moreover, BNP reduces renin secretion and is responsible for the inhibition of thirst, which induces vasodilation, and the decrease in blood volume and arterial pressure. Finally, BNP signaling is crucial for embryonic cardiac stem cell differentiation. It improves myocardial remodeling. The neurohormone is cleared from plasma mainly by binding to the natriuretic peptide receptor type C (NPR-C), which is expressed at high density in the glomerulus of the kidneys and its supportive vascular system. The proteases neprilysin and meprin-A are considered responsible for the catabolism of the above-mentioned peptides. The half-lives of BNP and NT-proBNP are 20 and 120 s, respectively.

As a biomarker, in the adult population, it has been thoroughly studied, and its clinical applications, mainly in cardiology and endocrinology, are well documented. Normal BNP reference values in healthy adults and reference intervals indicative of various pathological conditions have been determined in large multicenter studies, and therefore, it may be used as a tool for assessing response to treatment [2]. Further research in the pediatric age group is eagerly anticipated. Τhere has been growing interest concerning its clinical utility as a prognostic factor regarding the expected outcome for patients with conditions related to prematurity (HsPDA being a typical example [3], bronchopulmonary dysplasia [4], retinopathy of prematurity [5]), or conditions affecting older children (Kawasaki syndrome [6] and MIS-C [7]); reference intervals, though, for neonates, and children have not yet been established.

The aim of this systematic review and meta-analysis was a literature search of the detection of Nt-proBNP levels in healthy full-term neonates at birth using UCB samples and from serum collected during the first month of life (neonatal age). Furthermore, we aimed at determining the potential effect of demographic and perinatal factors on its reference range during this particular period of time.

## 2. Methods

### 2.1. Search Strategy

This systematic review followed the guidelines of the Preferred Reporting Items for Systematic Reviews and Meta-Analysis (PRISMA) statement [8].

We conducted a systematic review of the literature published as of the year 2000. The study consists exclusively of English-language articles. A scoping review framework that provided a systematic methodology using the electronic databases “Medline”, “Google Scholar”, and “Scopus” was undertaken. Expanded gray literature (including publications that were not reviewed by scientific journals) using appropriate MeSH terms was added to the searches of electronic databases.

### 2.2. Inclusion and Exclusion Criteria

The objective of this study was the detection of NT-proBNP levels in UCB and in the serum of healthy term newborn infants. Τerm birth was defined as between 37 and 41 weeks of gestation.

Studies that included preterm neonates (birth at <37 weeks of gestation), or neonates with congenital infections or defects, and papers mentioning neonatal or maternal perinatal morbidity were excluded from this review. Incomplete medical records were also excluded from the analysis. Congress abstracts, theses, dissertations, studies that did not involve human subjects, and papers that failed to meet the definition of original research did not meet the inclusion criteria of this study. Observational studies that determined UCB and serum ΝΤ-proBNP levels in healthy term neonates met the inclusion criteria/The control groups from studies that compared UCB and serum ΝΤ-proBNP levels in healthy term neonates and neonates with morbidities (type III fetal heart rate, neonatal stress, IUGR neonates, offsprings of women with type I diabetes, nonimmune hydrops, neonates with congenital heart disease, transient tachypnea of the newborn, respiratory distress syndrome) were also included in this study.

As far as the serum NT-proBNP values are concerned, the classification into subgroups according to the neonate’s day of life on which they were obtained posed a difficulty, since there was no accordance on the day of the one-month period that the biomarker was measured between the identified studies. In case multiple values were measured, those that included the first day of life were selected, since they represent the majority of the total values. Hence, those obtained on a later day of the neonate’s life were not included in the meta-analysis.

With regard to the biomarker’s concentration units used in the selected studies (ng/L, fmol/L, pg/mL), all measurements were converted into pg/mL in order to enable statistical data analysis.

### 2.3. Data Extraction

Literature search and data extraction were independently performed by 2 reviewers (E.C. and Z.I.). The abstracts were thoroughly evaluated for the selection of the appropriate studies. Τhe identified studies were properly tested for eligibility after screening titles and analyzing the abstracts for inclusion and exclusion criteria, resulting in an extensive assessment of existing literature. The full texts of potentially relevant studies were obtained and were evaluated after the authors, the year of publication, the study design, any demographic and perinatal factors involved, the biomarker’s values, and the main findings were taken into account.

### 2.4. Study Quality

For quality appraisal, the BIOCROSS evaluation tool was applied. BIOCROSS is a new study quality assessment tool, designed specifically to evaluate the reliability of observational studies employing biomarker data [9]. It facilitates a comprehensive review of a biomarker’s reference range that enables the evaluation of normal or abnormal medical conditions.

The BIOCROSS tool includes ten items divided into five domains: “Study Rationale”, “Design/Methods”, “Data Analysis”, “Data Interpretation”, and “Biomarker Measurement”. Study quality is low for scores less than or equal to 6, moderate between 7 and 12, and high if greater than 13.

### 2.5. Risk of Bias

Although the BIOCROSS evaluation tool has not been specifically designed to accurately calculate the potential risk of bias, it consists of several subsections that may serve as raters for its indirect estimation. The third item assesses the description of study population characteristics and the confounding factors that may influence the biomarker’s measurements. The fourth and fifth items assess the pertaining methods used to conduct statistical analysis and appraise the quality of the procedure used to measure the biomarker of interest.

### 2.6. Statistical Analysis

Statistical data processing for the meta-analysis was performed in the R programming software language (version 4.0.4) [10] within the Microsoft Windows environment using the package meta (version 4.18-0) [11,12]. For each parameter under investigation, a forest plot along with the results is presented. A funnel plot was also used to estimate the bias.

Although the selected studies provided little conformity in terms of the presentation of the results, the mean value and the standard deviation were required to conduct meta-analysis on arithmetic data. In cases where this was not possible, an estimation was made by calculating the median and the values of the 1st and 3rd quartiles according to Hozo et al. [13]. Moreover, in case the minimum and maximum values were available, an improved estimation, as proposed by Bland [14], was applied using the Deep Meta Tool, Version 1 [15]. Finally, if the 1st and 3rd quartile values were not available, the range rule was used to estimate the standard deviation value.

In the meta-analysis forest plots, the results are presented using two approaches: (a) the fixed-effects model and (b) the random-effects model. In the fixed-effects model, it is assumed that one phenomenon exists with a specific effect size that all involved studies are estimating. Thus, the weights assigned to studies are based on each one’s sample size, and large studies dominate the analysis results. The random-effects model estimates the mean value of numerous similar phenomena; thus, each study was considered to estimate a different effect size. In this approach, each effect reported by studies is used to calculate a mean effect and a standard deviation. In comparison to the fixed-effects model, in the random-effects model, the weights are more balanced, and large studies do not dominate the results. It is not possible to know if there is a single phenomenon and if it is affected by factors that change across studies (e.g., infant’s age); thus, both approaches are presented.

## 3. Results

The initial search was performed by analyzing the titles, abstracts, and keywords. A total of one hundred twenty-two (122) papers were retrieved from several bibliographic databases. Further abstract screening yielded seventy-four articles (74), which were found potentially appropriate to be included. After full-text evaluation, thirty-four articles (34) were eliminated, mainly due to a lack of conformity between the identified studies (Ν = 20) on the specific day of the one-month neonatal period that NT-proBNP levels were measured in serum. Following this, the selected studies underwent eligibility selection, and a number of them were removed from the analysis, primarily due to incomplete medical records as far as the measurement of the biomarker’s levels is concerned (N = 6), cases of non-English-language publications (Ν = 4) and issues concerning the type of study (abstracts, etc.) (N = 4). A PRISMA flow diagram is presented in Figure 1.

### 3.1. Participants

The selection process resulted in a final sample of forty (40) studies. A total of twenty-five (25) studies reported NT-proBNP concentrations in UCB samples of term neonates whose mothers did not exhibit any signs of perinatal morbidity. This sample consisted of 2232 term neonates. The rest of the studies, fifteen (15) in total, reported NT-proBNP concentrations in the serum of healthy neonates. This group included 828 neonates.

The meta-analysis included eighteen (18) studies concerning UCB samples with 1738 neonates as participants and fourteen (14) studies concerning serum samples collected during the neonatal period with 393 participants.

### 3.2. Umbilical Cord NT-proBNP in Term Neonates

The studies included in the systematic review are presented in Table 1 [16,17,18,19,20,21,22,23,24,25,26,27,28,29,30,31,32,33,34,35,36,37,38,39,40].

The NT-proBNP values reported in the selected studies were derived either from the umbilical vein (UV) (nine studies), the umbilical artery (UA) (five studies), or an unspecified umbilical blood sampling vessel (six studies). Regardless of the sampling site and the mode of delivery, NT-proBNP levels were 490 pg/mL on average (95% CI: 479–502 pg/mL) after applying the fixed-weight method and 838 pg/mL (95% CI: 676–999 pg/mL) after applying the random-weight model (shown in Figure 2a). The funnel plot, which is used as a method to assess the potential risk of publication bias is presented in Figure 2b.

Among the selected studies of our systematic review, the biomarker was measured in the UV in ten (10) of them and in the UA in eight (8) of them, while the UCB vessel was not specifically stated in nine (9) of them. A comparison between the biomarker’s levels in UA and UV was approached in only three (3) studies, which concluded that the sampling vessel does not particularly affect the measurements in a statistically significant manner, since only a modest disparity was observed between the two sampling sites, as shown by Kocylowski et al. [18] (UA vs. UV: (mean) 832 pg/mL vs. 831 pg/mL, *p* > 0.001), Bakker et al. [23] (UA vs. UV: (mean) 668 pg/mL vs. 669 pg/mL, *p* > 0.001) and Schwachtgen et al. [26] (UA vs. UV: *p* = 0.652). Contrarily, Blohm et al. [38] described a statistically significant difference between the aforementioned values (UA vs. UV: (median) 765.48 pg/mL vs. 816.45 pg/mL, *p* < 0.001). A meta-analysis was performed by combining data from available studies to produce a comparison between the biomarker’s values according to the chosen UCB vessel, regardless of the mode of delivery (shown in Appendix A). Between the different sampling sites significant heterogeneity was observed when the fixed-weight approach was applied, with lower NT-proBNP levels measured in UV blood–431 pg/mL—(95% CI: 417–446 pg/mL), whereas UA blood NT-proBNP values were 785 pg/mL (95% CI: 754–817 pg/mL) (*p* < 0.0001).

In thirteen (13) studies, a comparison between NT-proBNP levels according to the mode of delivery was attempted. Several of these studies just mentioned this comparison, and the exact values were omitted. Statistically significant increased levels in cesarean section (CS) as opposed to vaginal delivery (VD) were observed in the studies conducted by Kocylowski et al. [18] (CS vs. VD: (median) 630.3 pg/mL vs. 861 pg/mL, *p* > 0.001), Cardo et al. [17] (CS vs. VD: (mean) 662 pg/mL vs. 524 pg/mL, *p* > 0.001), Fortunato et al. [19] (CS vs. VD: (mean) 730 pg/mL vs. 500 pg/mL, *p* < 0.001), in contrast with studies conducted by Won Joon Seong et al. [16] (CS vs. VD: (mean) 801.9 pg/mL vs. 724.3 pg/mL, *p* = 0.572), Bar-Oz et al. [21] (N/A), Bakker et al. [23] (Ν/A), Schwachtgen et al. [26] (CS vs. VD: *p* = 0.229), Rouatbi et al. [29] (CS vs. VD: *p* = 0.336), Iacovidou et al. (Ν/A) [30], Halse et al. [31] (CS vs. VD: *p* = 0.09), Russell et al. [33] (N/A), Girsen et al. [35] (N/A), and Ersoy et al. [40] (*p* = 0.77). Among the articles that provided relevant data, we performed the meta-analysis based on three (3) studies. The NT-proBNP concentration in CS is estimated at 954 pg/mL (95% CI: 826–1082 pg/mL) when the fixed model was applied. Lower levels, i.e., −798 pg/mL—(95% CI: 688–908pg/mL, *p* = 0.04), were measured in VD (shown in Appendix A), displaying a relatively low heterogeneity rate (i^2^ = 60%) and bias.

In seven (7) of the identified studies, a potential correlation between the UCB NT-proBNP levels of term neonates and their mother’s serum NT-proBNP levels prenatally was studied. Specifically, elevated UCB NT-proBNP concentrations in comparison to maternal blood NT-proBNP levels were noted by Bar-oz et al. [21] (UCB NT-proBNP vs. maternal blood NT- proBNP: (mean) 612.2 pg/mL vs. 89.7 pg/mL, *p* < 0.001), Hammerer-Lercher et al. [22] (UCB NT-proBNP vs. maternal blood NT-probnp: (median) 553.4 pg/mL vs. 45.5 pg/mL, *p* < 0.001), Bakker et al. (N/A) [23], Lechner et al. [24] (UCB NT-proBNP vs. maternal blood NT-proBNP: (median) 626 pg/mL vs. 66 pg/mL, *p* < 0.001), Iacovidou et al. [30] (*p* = 0.002), Blohm et al. [38] (UV NT-proBNP vs. maternal blood NT- proBNP: (median) 816.45 pg/mL vs. 44.4 pg/mL, *p* < 0.001), and Blohm et al. [39] (UV NT-proBNP vs. maternal blood NT- proBNP: (mean) 1228.94 pg/mL vs. 71.48 pg/mL, *p* = 0.000).

Furthermore, a thorough evaluation of the correlation between the infant’s sex and the biomarker’s UCB concentrations was attempted. No significant sex-dependent correlation was determined among the entire studies included in the review.

### 3.3. Serum NT-proBNP in Term Neonates

The studies that compose the present review are depicted in Table 2 [21,26,41,42,43,44,45,46,47,48,49,50,51,52].

Τhe identified studies displayed little accordance on a specific day, through the 30-day neonatal period, that the biomarker’s serum levels were measured. In case multiple values on separate days were available, those that included the first or second day of life were selected, since they represent the majority of the total values. Statistical data from fourteen (14) studies were available for processing. The forest plot is depicted in Figure 3a. According to the fixed-effects model, the NT-proBNP serum levels were 1341 pg/mL (95% CI: 1286–1397 pg/mL), while the random-effects model showed an almost 3-fold elevation of levels at 3176 pg/mL (95% CI: 2049–4303 pg/mL). The associated funnel plot (shown in Figure 3b) is indicative of the high bias rate that characterizes the studies. The comparison between cumulative NT-proBNP values in serum and UCB demonstrates a relatively higher serum concentration (*p* < 0.0001).

Another observation is that NT-proBNP levels gradually decrease as the ages of the neonate’s advance. Several studies have reported a negative correlation between NT-proBNP values and age such as the studies of Rauh et al. [41] (r = −0.45, *p* < 0.001), Li et al. [46] (r = −0.23, *p* < 0.05), and Deng et al. [48] (r = −0.706, *p* = 0.000). Nir et al. [42] reported significantly higher levels of the biomarker during the neonate’s first days of life as opposed to later days (*p* < 0.05), while infants between 4 months and 15 years old did not exhibit a remarkably wide range of values (regression analysis, r^2^ = 0.18). Soldin et al. study [43] reported that during the neonatal period, NT-proBNP levels were elevated up to 20 times in comparison to infancy, even though the study population was limited in size. Additionally, NT-proBNP levels slowly decrease and gradually approach the adult reference range, as demonstrated in the studies of Rauh et al. [41], Koch et al. [53], and Deng et al. [48]. Schwachtgen et al. [26], on the other hand, observed a rapid NT-proBNP increase during the first 2 days of life, when compared to their respective UCB levels, which is succeeded by a rapid decrease during the first year of life, that persists until normal values are achieved approximately at early adulthood. Similar results appeared in the studies of Caselli et al. [44] and Zhu et al. [49]. Markovic-Sovtic et al. [52] claimed that elevated values are measured during the 1st week of life in comparison to those obtained during the rest of the 1st month (*p* < 0.001). In contrast to the aforementioned studies, Mir et al. [54] reported no statistically significant difference according to the infant’s day of life as far as the biomarker’s concentration is concerned.

None of the elected studies proved that the biomarker’s values are affected by the infant’s sex, as shown in the following studies.

### 3.4. Quality of Studies

As far as studies about the UCB NT-proBNP concentration are concerned, four (4) out of twenty-five (25) were characterized as of “low quality”, twenty (20) of moderate quality, and one (1) met the criteria of the high-quality study, according to the BIOCROSS evaluation tool (shown in Figure 4a).

Articles that obtained low scores displayed underlying errors or defects particularly concerning the description of exclusion criteria, specimen characteristics, assay methods, laboratory measurements, and biomarker modeling. Specifically, only nine (9) out of the twenty-two (22) studies mentioned the quality control procedures of biomarker measurement (e.g., inter- and intra-coefficient of variation). There are important limitations concerning the reproducibility assessment performed for evaluating the biomarker’s stability (2/22), as well as the citation of potential technical errors in measurements (0/22). On the contrary, the effect of potential confounding factors was relatively well established, since only two (2) of the studies presented a lack of such assessment.

As far as the fifteen (15) studies that referred to serum NT-proBNP levels of healthy term neonates during the neonatal period are concerned, five (5) of them were of low quality, and the rest were characterized as of moderate quality (shown in Figure 4b). Studies on the biomarker’s UCB levels display similar limitations.

## 4. Discussion

The present systematic review and meta-analysis aimed at studying the UCB and serum ΝΤ-proBNP levels in healthy term neonates during their first month of life.

As embryonic development progresses, BNP follows a different metabolic pathway in comparison to ANP, whose UV levels appear elevated when compared to those that derive from the UA, implying a potential endogenous placental production, accounting for a role in fetoplacental hemodynamics [55]. On the contrary, BNP, and consequently, NT-proBNP concentrations, do not display a decrease after passing through the placenta, attributable to a lower NPR-C receptor affinity and a higher neutral endopeptidase affinity [56,57]. Moreover, it appears that the secretion of NT-proBNP detected in the amniotic fluid derives primarily from embryonic cardiomyocytes [55,58]. Suga et al. [57] and Carvajal et al. [59], however, suggested that it derives from the amniotic fluid, mediated by paracrine mechanisms of myometrial cells, before ultimately entering the fetal circulation. NT-proBNP levels are significantly elevated in healthy pregnancies until mid-pregnancy and tend to reach a plateau at around 34 weeks of gestation. Specifically, the study of Merz et al. [58] reported that the mean NT-proBNP concentration is 1998 pg/mL between 20 and 34 weeks of gestation, while the Carvajal et al. [60] study calculated the same concentration to be at 1782 pg/mL.

Blohm et al. [38] reported that UCB and amniotic fluid levels of the biomarker at birth do not exhibit a remarkable correlation (amniotic fluid NT-proBNP vs. UV NT-proBNP: (mean) 816.45 pg/mL vs. 72.03 pg/mL, *p* < 0.01), which was also supported by the study of Esroy et al. [40] (Spearman’s r = 0.2, *p* = 0.07), and results from the crucial hemodynamic changes that occur during the transition from intrauterine to extrauterine life. The study of Miyoshi et al. [61], however, proved a relatively weak, yet statistically significant, correlation between UCB and amniotic fluid values of the biomarker at birth (Spearman’s r = 0.51, *p* < 0.01), notwithstanding the detection of 22-fold higher UV NT-proBNP levels compared to the respective amniotic fluid levels.

Although there has been growing interest concerning the clinical utility of NT-proBNP as a prognostic and diagnostic tool regarding neonatal morbidity, a robust assessment of the levels expected in neonates is yet to be performed, since it is an essential prerequisite in clinical practice. In the present meta-analysis, the mean value of NT-proBNP was calculated at 490 pg/mL (95% CI: 479–502 pg/mL) using the fixed-effects model and at 838 pg/mL (95% CI: 676–999 pg/mL) according to the random-effects model. Significant heterogeneity was documented among the selected studies, since they referred to study populations with distinct racial characteristics, they presented low representativeness that could be attributed to specific study design requirements, and they had a limited size population. No detectable difference in values was observed with the mode of delivery, the infant’s sex, or the UCB vessel.

A few studies undertook a comparison between the biomarker’s levels according to the UCB sampling vessel. No statistically significant difference was reported in three (3) of them [18,23,26], as opposed to the study of Blohm et al. [38], which detected elevated UV NT-proBNP values in comparison to the respective UA values. The present meta-analysis, which includes data from twelve (12) studies, depicts a minor yet statistically significant difference in favor of the UA, in comparison to the UV sampling site, according to the fixed-effects model (431 pg/mL, 95% CI: 417–446 pg/mL vs. 785 pg/mL, 95% CI: 754–817 pg/mL, *p* < 0.001).

Concerning the mode of delivery, elevated NT-proBNP levels were noted in the case of CS in comparison to VD in three (3) of the selected studies [17,18,19], whereas a total of ten (10) studies observed no such difference [16,21,23,26,29,30,31,33,35,40]. Our meta-analysis (which included three (3) studies that met the inclusion criteria) is consistent with the findings of the first group of researchers, since we concluded that the values measured were slightly elevated in the case of CS (CS vs. VD: 954 pg/mL (95% CI: 826–1082 pg/mL) vs. 798 pg/mL, (95% CI: 688–908 pg/mL, *p* < 0.001). The limited number of studies that met the inclusion criteria accounts for the relatively poor reliability of the aforementioned result. It is safe to assume that no significant correlation actually exists between CS, mode of delivery, and high levels of neonatal stress or cardiac overload, which are considered responsible for the increased production of NT-proBNP at birth. A noteworthy fact is that a remarkable difference is observed neither in the case of emergency CS, nor in the case of emergency spontaneous VD [19,31], which denotes that delivery constitutes a considerably stressful procedure for the fetus, and consequently, NT-proBNP levels fail to serve as an independent discriminant biomarker of stress [29].

None of the selected studies proved that the biomarker’s values are affected by the infant’s sex. This remains the case until late childhood, since in adolescence, the substantially higher levels of NT-proBNP in female subjects can be attributed to the inverse regulatory association between testosterone and NT-proBNP, suggesting that androgens modulate sex differences notable in the secretion of natriuretic peptides (NPs) [62,63].

A remarkable discrepancy was confirmed between NT-proBNP concentrations that derive from UCB and maternal serum samples during labor. More specifically, five (5) studies that attempted a comparison [21,22,24,38,39] provided the observation that NT-proBNP UCB values presented a 15-fold increase according to the fixed model method (95% CI: 14–16, *p* < 0.01) (shown in Appendix A). NT-proBNP’s production, secretion, and clearance of NPs are regulated independently in the neonatal and maternal circulation.

As described in the present meta-analysis, the serum NT-proBNP levels in healthy term infants during the neonatal period were 1341 pg/mL (95% CI: 1286–1397 pg/mL) and 3176 pg/mL (95% CI: 2049–4303 pg/mL) according to the fixed-effects model and the random-effects model, respectively. High levels of heterogeneity across studies were also identified in this case. Higher serum NT-proBNP levels are observed in comparison to UCB NT-proBNP levels during the neonatal period (*p* < 0.0001). This scientific obstacle, which is confirmed in literature references [21,22,25,26,50,64], derives from multiple pathophysiological mechanisms, such as the placenta influence on BNP clearance [64], the newborn’s water loss increase postnatally, the renal function immaturity [65], and the significant changes that occur due to the gradual functional closure of ductus venosus, foramen ovale, and ductus arteriosus during the first hours of life of term neonates [66].

The postnatal profound hemodynamic changes and, specifically, the pulmonary perfusion and the systemic afterload increase are responsible for the high right ventricular cardiac volume and cardiac output, which leads to a BNP gene expression stimulation. The procedure is mainly mediated by mitogen-activated protein kinases and, specifically, the extracellular signal-regulated kinase, which is activated as a response to the stimulus of pressure overload on the cardiac myocytes [67]. Consequently, the NT-proBNP concentration is relatively high in the first week and then gradually decreases significantly. The decrease in plasma NT-proBNP can be attributed to the diuresis that goes along with renal maturation, the suppression of renin-aldosterone-angiotensin I and II system activity, and the neonate’s water loss increase postnatally [68]. The gradual serum level decrease in the biomarker continues until the first months of life and remains considerably stable from infancy to the beginning of adolescence [25,26,41,43,44,45,46,47,48,49,52,54,69].

A number of practical scientific obstacles were encountered throughout the process of conducting the present systematic review and meta-analysis; thus, several important limitations are mentioned hereby. To begin with, employing the cross-sectional survey design results in obvious study limitations. Then, vascular punctures in healthy newborns, especially if performed without indication, are associated with higher levels of neonatal stress and perinatal morbidity [70]. Therefore, there has been a tendency to avoid any purposeless invasive procedures performed in neonates during the past decades, which in turn causes difficulties in conducting studies that aim to establish intervals for the biomarker levels in blood samples of healthy neonates, since the majority only include NT-proBNP values that refer to neonates with morbidities. Third, various clinical laboratory NT-proBNP measurement methods are reported. The different kinds of immunoassays lack necessary standardization, and the results that derive from different kits are not sufficiently comparable [71]. The inter-assay discrepancy may also be attributed to the heterogeneity of antibodies that are used to detect different NT-proBNP fragments, as well as the fact that little conformity exists between different analyzer kit labels concerning the technical characteristics of the in vitro determination of the biomarker’s concentrations [72,73]. This could probably provide an explanation for the high levels of heterogeneity that were observed among the selected studies. Nevertheless, in order to ensure satisfactory reliability, a large number of studies have been examined. Moreover, several studies were considered to be of low quality according to the BIOCROSS evaluation tool, which results in a potential publication bias. Large multicenter studies in populations of healthy term neonates with distinct demographic characteristics are required in order to determine intervals for NT-proBNP by analyzing UCB and serum samples during the neonatal period. Last, further investigation on the potential effect of perinatal and demographic factors on the biomarker’s range is undoubtedly necessary.

## 5. Conclusions

The current study pioneers in analyzing and introducing reliable intervals for NT-proBNP in UCB and healthy term neonates’ serum during the neonatal period. Furthermore, an identification of presumable correlations between the biomarker’s levels and certain variables of the perinatal period or neonatal demographic characteristics was accomplished. It is common knowledge that infant reference intervals are rather difficult to address in modern healthcare, since access to blood samples from healthy neonates is limited by ethical and practical constraints. The conduction of reliable large-scale studies with a representative sample of participants is greatly anticipated in an attempt to establish accurate reference intervals specific to the neonatal population.

## Figures and Tables

**Figure 1 diagnostics-12-01416-f001:**
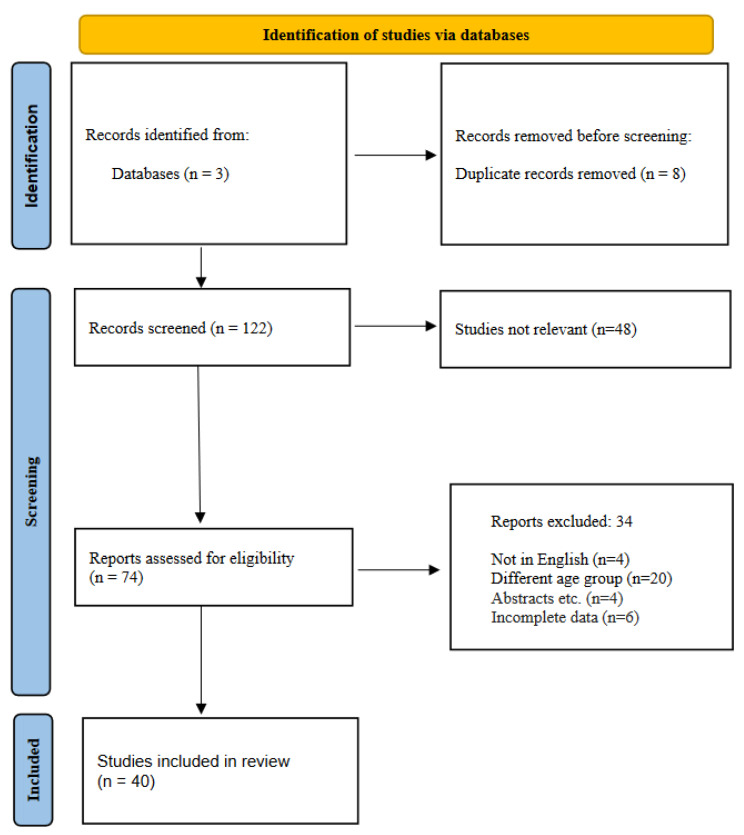
The PRISMA flow diagram of the study selection process.

**Figure 2 diagnostics-12-01416-f002:**
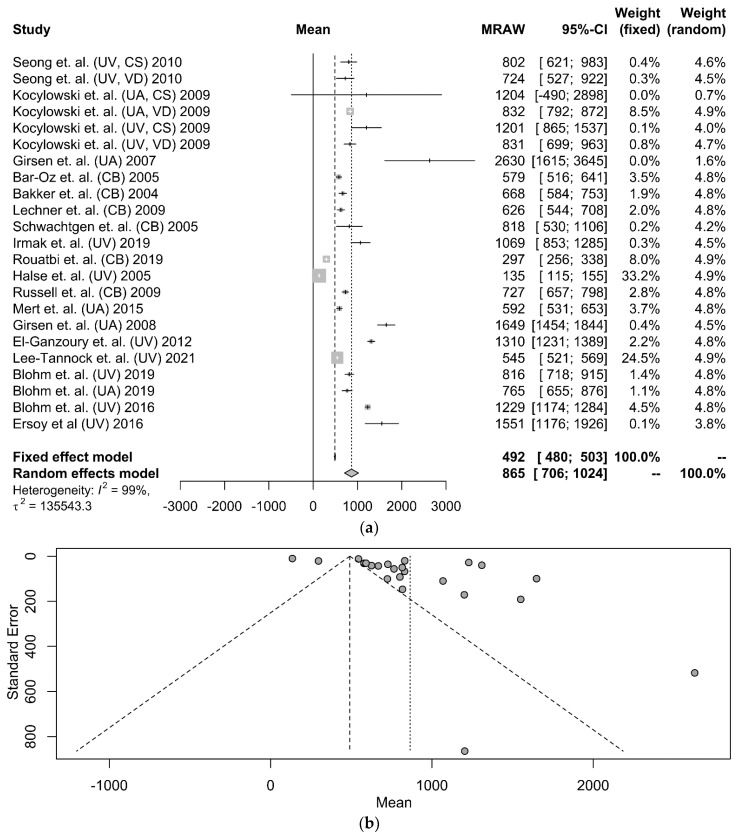
(**a**) A forest plot that presents the results of the meta-analysis: umbilical cord NT-proBNP values of term infants using the random model and the fixed model, respectively. The first column refers to the first author of the study, the year of publication, and the sampling vessel. The second column represents the mean value and the standard deviation graph. The next 3 columns show the mean value and the 95% confidence interval. The remaining 2 columns present the fixed model and random model weights [16,18,20,21,23,24,26,27,29,31,33,34,35,36,37,38,39,40]. (**b**) The corresponding funnel plot is presented at the bottom. (UV: umbilical vein, UA: umbilical artery, CB: cord blood, CS: cesarean section, VD: vaginal delivery).

**Figure 3 diagnostics-12-01416-f003:**
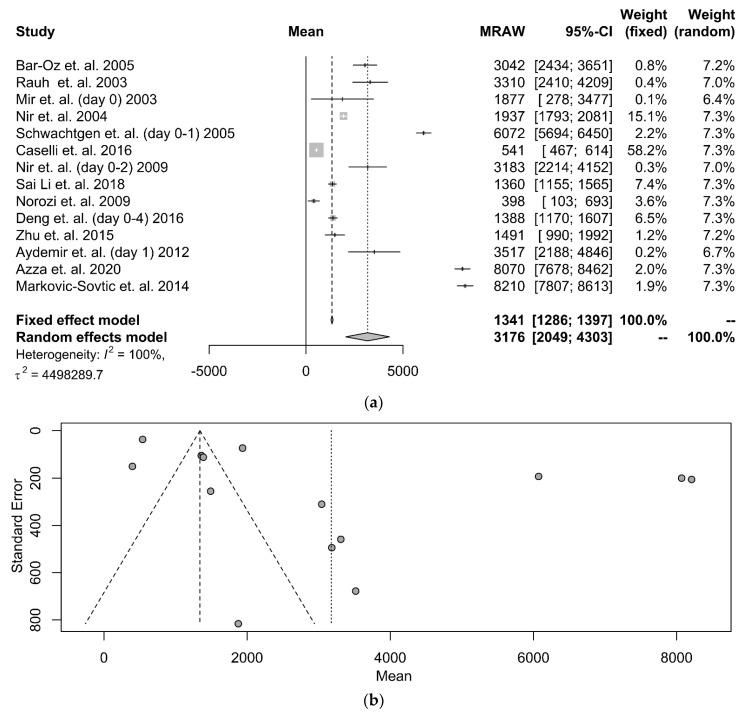
(**a**) A forest plot that presents the meta-analysis of the NT-proBNP values of healthy term neonates during the neonatal period, after the application of the random or the fixed model. The first column refers to the first author, the year of publication, and the day of neonates’ life on which the measurement occurred (in parentheses). The second column represents the mean value and the standard deviation graph. The next 3 columns show the mean value and the 95% confidence interval. The remaining 2 columns present the fixed model and random model weights [21,25,26,41,42,44,45,46,47,48,49,50,51,52]. (**b**) A funnel plot that presents the meta-analysis of NT-proBNP values in healthy term neonates at the bottom.

**Figure 4 diagnostics-12-01416-f004:**
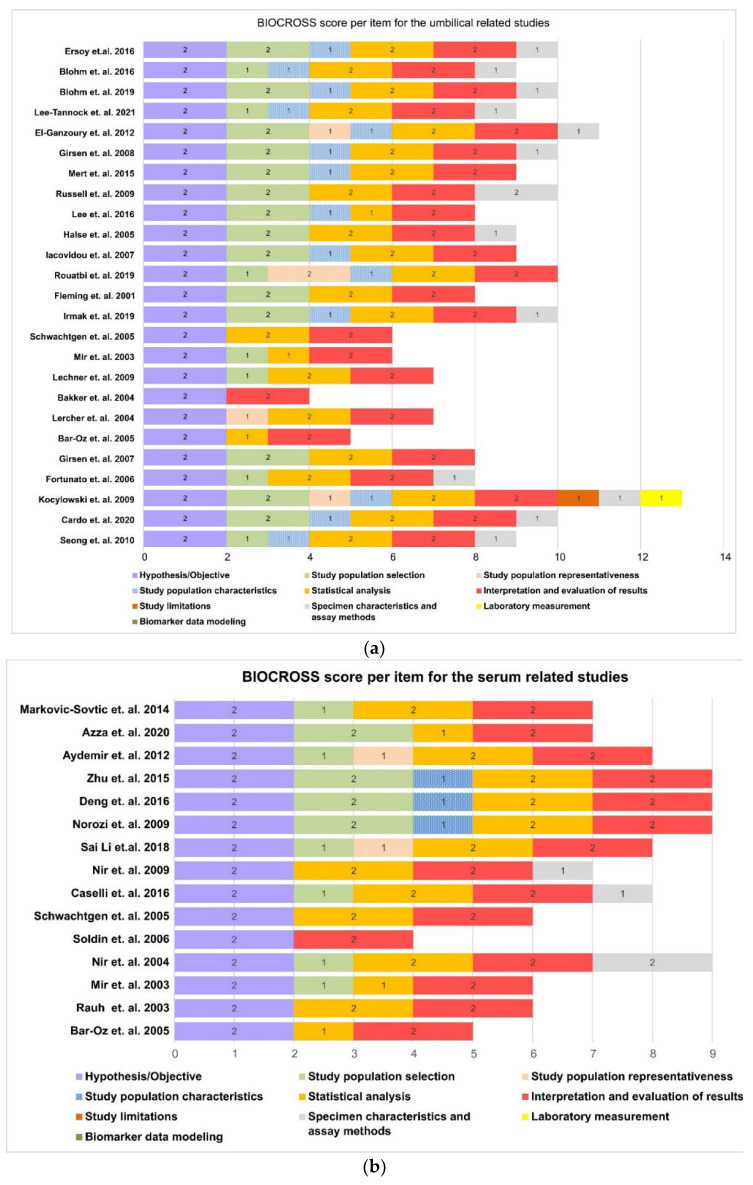
(**a**) Biocross evaluation tool for the assessment of the umbilical cord studies [16,17,18,19,20,21,22,23,24,25,26,27,28,29,30,31,32,33,34,35,36,37,38,39,40]. (**b**) Biocross evaluation tool for the assessment of the serum studies [21,25,26,41,42,43,44,45,46,47,48,49,50,51,52].

**Table 1 diagnostics-12-01416-t001:** Characteristics and results of included studies that report on the values of NT-proBNP in umbilical cord of healthy term neonates (N = 25). N/A: not applicable.

First Author	Year	Source	Mode of Delivery	N	Kit	NT-proBNP (pg/mL)
Won Joon Seong [16]	2010	Umbilical vein	CS	34	Modular Analytics	Mean: 801.9 SD: ±537.7
		Umbilical vein	VD	29		Mean: 724.3 SD: ±542.4
Leire Cardo [17]	2020	Cord blood	CS	92	Cobas 6000	Mean: 662 SD: -
			VD	177		Mean: 524 SD: -
Rafal Kocylowski [18]	2009	Umbilical artery	CS	2	Elecsys 2010	Mean: 1204 SD: ±1222.6
		Umbilical artery	VD	154		Mean: 832 SD: ±253.1
		Umbilical vein	CS	51		Mean: 1201 SD: ±1222.6
		Umbilical vein	VD	229		Mean: 831 SD: ±1022.4
Giuliana Fortunato [19]	2006	Cord blood	N/A	87	Elecsys 2010	Mean: 548.2 SD: -
Anna Girsen [20]	2007	Umbilical artery	N/A	49	N/A	Median: 2630.1 Range: 1319–15814
Benjamin Bar-Oz [21]	2005	Cord blood	N/A	122	Elecsys 2010	Mean: 578.8 SD: ±351.3
Angelika Lercher [22]	2005	Umbilical vein	N/A	42	Elecsys 2010	Median: 553.4 SD:-
Jaap Bakker [23]	2004	Cord blood	N/A	71	Elecsys 2010	Μean: 668.1 SD: ±363.5
Evelyn Lechner [24]	2009	Cord blood	N/A	200	Elecsys 2010	Median: 626 Range: 153–2518
Thomas Mir [25]	2003	Umbilical vein	VD	37	Biomedica	Mean: 1691 SD: -
Lynn Schwachtgen [26]	2005	Cord blood	N/A	62	Elecsys 2010	Mean: 818 SD: ±281–2595
Kübra Irmak [27]	2019	Umbilical vein	CS	43	CK-E10219	Mean: 1069 SD: ±721
Sean Martin Fleming [28]	2001	Umbilical artery	N/A	16	Biomedica	Mean: 1885.5 SD: -
Hatem Rouatbi [29]	2019	Cord blood	N/A	169	Elecsys 2010	Mean: 296.87 SD: ±273.8
Nicoletta Iacovidou [30]	2007	Cord blood	N/A	20	Biomedica	Mean: 8032.6 SD: -
Karen Halse [31]	2005	Umbilical vein	N/A	22	N/A	Median: 135,2 Range: 76.1–270.6
Seung Lee [32]	2016	Umbilical vein	N/A	23	N/A	Median: 633.2 SD:-
Noirin Russell [33]	2009	Cord blood	N/A	39	Elecsys 2010	Median: 727.3 Range: 50.7–1403.8
Mustafa Mert [34]	2015	Umbilical artery	N/A	58	ECLIA	Mean: 592 SD: ±236.8
Anna Girsen [35]	2008	Umbilical artery	N/A	60	N/A	Median: 1649,1 Range: 414.3–3501.1
Mona El-Ganzoury [36]	2012	Umbilical vein	N/A	30	R & D Systems	Median: 1310.5 Range: 870.9–1750.2
Alison Lee-Tannock [37]	2021	Umbilical vein	N/A	78	Vitros 5600	Median: 545 Range: 368–793
Martin Blohm [38]	2019	Umbilical vein	N/A	86	Elecsys 2010	Median: 816.45 5th–95th Percentile: 369.62–2233.15
		Umbilical artery	N/A	48		Median: 765.48 5th–95th Percentile: 259.84–1819.5
Martin Blohm [39]	2016	Umbilical vein	N/A	66	Elecsys 2010	Mean: 1228.94 25th–75th Percentile: 708–1617.5
Ali Ersoy [40]	2016	Umbilical vein	N/A	36	Elisa kit (USCNK, Wuhan, Hubei)	Mean: 1551.18 SD: ±1148.9

**Table 2 diagnostics-12-01416-t002:** Characteristics and results of included studies that report on the values of NT-proBNP in serum of healthy term neonates (N = 15). N/A: NOT applicable.

First Author	Year	Age Range	N	Kit	NT-proBNP (pg/mL)
Benjamin Bar-Oz [21]	2005	Day 1–4	33	Elecsys 2010	Mean: 3042.4 SD: ±1783.2
Manfred Rauh [41]	2003	Day 0–12	13	Elecsys 2010	Range: 1121–7740
Thomas Mir [25]	2003	Day 0	14	Biomedica	Mean: 1877.1 SD: ±490.4–6586.8
		Day 1	18		Mean: 5419,9 SD: ±2147.6–10,755.3
		Day 2	12		Mean: 2703 SD: -
		Day 3	16		Mean: 2064.8 SD:
		Day 4	18		Mean: 2083.6 SD: -
		Day 5–7	16		Mean: 1724.9 SD: -
		Day 8–14	10		Mean: 1437.4 SD:-
		Day 15–30	12		Mean: 1830 SD: -
Amiram Nir [42]	2004	Day 1–5	20	Elecsys 2010	Mean: 1937 SD: ±328
Steven Soldin [43]	2006	Day 0–30	99	Dade R X L	97th: 31832.5
Lynn Schwachtgen [26]	2005	Day 0–1	8	Elecsys 2010	Mean: 6072 SD: ±546
		Day 2–3	40		Mean: 2972 SD: ±1808
		Day 4–8	11		Mean: 1731 SD: ±1236
Chiara Caselli [44]	2016	Day 0–30	24	Elecsys 2010	Median: 504.3 Percentiles 25th–75th: 211.1–942.7
Amiram Nir [45]	2009	Day 0–2	43	Elecsys 2010	Median: 3183 Range: 260–13,224
		Day 3–11	84		Median: 2210 Range: 28–7.250
Sai Li [46]	2018	Day 0–30	80	VIDAS	Median: 1360 Range: 250–3987
Kambiz Norozi [47]	2009	Day 0–30	8	Elecsys 2010	Mean: 398 SD: ±425
Menghong Deng [48]	2016	Day 0–4	N/A	YZB/CAN 1794-2008	Median: 1388.5 Range: 750–4615
		Day 5–15	Ν/A		Median = 640.8 Range 515–850
		Day 16–28	N/A		Median = 412.7 Range 341–462
Rui Zhu [49]	2015	Day 0–3	63	Elecsys 2010	Median: 1491 Percentiles 5th–95th: 499.5–8607.3
Ozger Aydemir [50]	2012	6th hour of life	33	Immulite 2000	Median: 3509 Min–Max: 1174–35,000
		Day 1	33		Median: 3517 Min–Max: 875–16,452
		Day 3	33		Median: 1211 Min–Max: 377–2774
		Day 5	33		Median: 808 Min–Max: 440–2284
Azza Ahmed [51]	2020	Day 0–30	26	Abnova	Mean: 8070 SD: ±1020
Gordana Marcovic [52]	2014	Day 1–28	28	Elecsys 2010	Mean: 8210 Range: 6.110–10.460

## Data Availability

All data generated or analyzed during this study are included in this article and its Appendix A. Further inquiries can be directed to the corresponding author.

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
