# Peer review of "NT-proBNP Concentrations in the Umbilical Cord and Serum of Term Neonates: A Systematic Review and Meta-Analysis"

_diagnostics, 2022, doi:10.3390/diagnostics12061416_

Round 1

Reviewer 1 Report

In the manuscript by Christou et al., the authors present a systematic review and meta-analysis of the current bibliography regarding the normal NT-proBNP concentrations in the umbilical cord and serum of term neonates. The authors claim that this is the first time that such a study has been conducted. Based on this, the results of the present study could be of potential interest. However, several points need further investigation: 

1. The Introduction section is quite brief and needs to be further expanded in order to help the reader understand the role of BNP in health and disease and the significance and potential of NT-proBNP as a biomarker.

2. Figure 1 needs to be corrected since paragraph symbols appear.

3. The authors have analyzed the results using and showing each time the fixed weight method and the random weight model. These two models give significantly different results. The authors should discuss the observed differences between the results of each method, as well as the difference between the two methods themselves.

4. In the discussion section, the authors presenthe serum NT-proBNP levels in healthy term infants during the neonatal period. However, especially regarding the random model there is a high deviation between recorded  values. Do the authors believe it would be safe to use of these values as reference for normal NT-proBNP levels?

5. Discussion should be more concise and targeted.

Reviewer 2 Report

In this systematic review authors evaluated the studies available in literature regarding the role of NT-proBNP in UCB samples, and from serum collected during the first month of life of healthy term neonates. They found that NT-proBNP levels were generally higher in UCB compared to serum samples. Moreover, authors also showed the reference intervals of NT-proBNP in UCB and in serum.

This manuscript is clear and generally well written. Morever, this data are very useful and suggest a possible use of NT-proBNP as marker for the evaluation of potential adverse effects of perinatal factors. Thus, I have only the following comments: 

  • pharagraph sign in figure 1 must be removed
  • an accurate revision of typing error is highly suggested

Round 2

Reviewer 1 Report

I have no further comments.